# Statistical Modelling of the Fatigue Bending Strength of Norway Spruce Wood

**DOI:** 10.3390/ma15020536

**Published:** 2022-01-11

**Authors:** Jernej Klemenc, Gorazd Fajdiga

**Affiliations:** 1Faculty of Mechanical Engineering, University of Ljubljana, Aškerčeva 6, 1000 Ljubljana, Slovenia; Jernej.klemenc@fs.uni-lj.si; 2Department of Wood Science and Technology, Biotechnical Faculty, University of Ljubljana, Jamnikarjeva 101, 1000 Ljubljana, Slovenia

**Keywords:** spruce wood, wood density, static material characteristics, fatigue strength, Weibull’s probability density function, augmented inverse power law

## Abstract

When wood is used as a structural material, the fact that it is a highly inhomogeneous material, which significantly affects its static and fatigue properties, presents a major challenge to engineers. In this paper, a novel approach to modelling the fatigue-life properties of wood is presented. In the model, the common inverse-power-law relationship between the structural amplitude loads and the corresponding number of load cycles to failure is augmented with the influence of the wood’s mass density, the loading direction and the processing lot. The model is based on the two-parametric conditional Weibull’s probability density function with a constant shape parameter and a scale parameter that is a function of the previously mentioned parameters. The proposed approach was validated using the example of experimental static and fatigue-strength data from spruce beams. It turned out that the newly presented model is capable of adequately replicating the spruce’s S-N curves with a scatter, despite the relatively scarce amount of experimental data, which came from different production lots that were loaded in different directions and had a significant variation in density. Based on the experimental data, the statistical model predicts that the lower density wood has better fatigue strength.

## 1. Introduction

In recent years, wood has become an increasingly interesting alternative to steel and concrete due to greater environmental awareness and the general trend towards sustainable construction materials. There are many studies that have investigated different materials and manufacturing processes to develop highly engineered wood products to achieve an increased load-bearing capacity and stiffness [1,2,3,4,5,6]. Products made from biological materials such as wood often exhibit complex mechanical behaviour. Although such materials have been used for thousands of years, their mechanical behaviour is not yet fully understood. Their properties often vary from sample to sample and exhibit non-linear mechanical behaviour when subjected to large loads.

Spruce is often used in the construction industry, either in the form of load-bearing axial or bending beams, or as the basic building material of various carpentry products, such as door and window frames. In most construction applications, the wooden structural elements are subjected to quasi-static loading with a small change in the load over time. In such applications it is the static strength of the wood that determines the load-carrying capacity of the structure. However, in some applications, when the structural elements are frequently exposed to unfriendly environmental conditions (e.g., windy locations) or a dynamic environment (e.g., vibration or repeated mechanical impact loads), the structure might fail due to the fatigue process. The fatigue strength of wood-based materials is lower than the corresponding ultimate static strength [7,8,9,10,11], which is typical for the vast majority of structural materials. Consequently, to enable the design of structural elements, the static and fatigue strengths of wood must be known. One of the first studies related to the fatigue of wood was carried out by Wood [12], who presented the so-called “Madison curve” as the relationship between the stress and the load duration. In recent years, much research has been conducted to explain the fatigue phenomena with respect to wood [8,10,11,13,14,15,16].

Unlike structural metals or polymers, wood has a very inhomogeneous structure. It depends on the growth conditions, the orientation of the cut elements, the relative humidity (reflected in the mass density of the wood), knots, resin channels, fibre deviations, reaction wood, etc. [8,9,10]. This means that a significant scatter of the strength characteristics is expected for each type of wood. Spruce, which was the subject of our research, is not an exception. During the research of its properties, static and fatigue bending tests were performed on beam-like specimens with the same geometry, but different properties (i.e., density, processing lot and orientation of the load with respect to the growth rings). Since a characterisation of the static properties of the material was previously published by Fajdiga et al. [17,18,19,20], the main focus in this paper is an evaluation of the fatigue properties of spruce. Nevertheless, a statistical analysis is also performed for the static data to show the difference between the static and fatigue behaviours of the wood.

The main problem that needs to be addressed when characterising the fatigue properties of inhomogeneous materials with significant variations of the structural characteristics is the scatter of the fatigue-life data. This means that not only the trend of the fatigue-life curve needs to be estimated, but also its scatter. Due to the many influencing factors that have an effect on the durability of spruce, the scatter of the fatigue-life curve (i.e., a diagram of the number of load cycles to failure *N* vs. the loading levels *S*) can span more than an order of magnitude along the *N* axis of the S-N curve. An overview of the statistical methods for a data analysis of this kind is presented in the book of Nelson [21]. According to the ASTM E 739-91 standard [22], the fatigue-life curve and its scatter in the high-cycle fatigue domain are modelled by combining the log-normal probability density function (PDF) *f*(*N*|*S*) of the number of load cycles to failure *N* with an inverse power-law relationship between the number of load cycles to failure *N* and the loading level *S* as follows:

A linear-regression model is first set-up for the log(*N*) vs. log(*S*) relationship on the basis of *n* experimental data points **S** = {(*S_i_*,*N_i_*); *i* = 1,…, *n*}:
(1)$$\log\left(\hat{N}\right)=a_{0}+a_{1}\cdot \log\left(S\right);a_{0}>0,a_{1}<0$$
The linear-regression model from Equation (1) represents the mean value *μ* of the conditional log normal PDF *f*(*N*|*S*). Its standard deviation is calculated from the deviations between the logarithmic values of the measured *N_i_* and the predicted ${\hat{N}}_{i}$ load cycles:(2)$$\sigma =\sqrt{\frac{1}{n-1}\cdot \sum_{i=1}^{n}{\left[\log\left({\hat{N}}_{i}\right)-\log\left(N_{i}\right)\right]}^{2}}$$

The approach of the ASTM E 739-91 standard [22] works well if all the fatigue-life data from the sample set **S** = {(*S_i_*,*N_i_*); *i* =1,…, *n*} represent the fatigue failures. However, if the sample set also consists of run-out experiments, this procedure cannot be directly applied to the complete dataset. Run-out experiments are common during fatigue testing, because the tests are usually terminated if no fatigue failure occurs before a predefined number of loading cycles. To set-up the fatigue-life curve model from Equations (1) and (2), the run-out data should either be discarded or they should be considered using the approach that was proposed by Pascual and Meeker [23]. Following their procedure, the parameters *a*_0_, *a*_1_ and *σ* from Equations (1) and (2) are estimated using an enhanced maximum-likelihood function, in which the fatigue failures are considered via the PDF *f*(*N*|*S*) and the run-outs are considered via the corresponding cumulative probability function *F*(*N*|*S*). This is a problem if the PDF *f*(*N*|*S*) has a log-normal form, since its cumulative probability function *F*(*N*|*S*) cannot be analytically determined. The solution is to replace the log-normal PDF *f*(*N*|*S*) with a conditional Weibull’s PDF [24].
(3)$$f\left(N|S\right)=\frac{\beta }{\eta }\cdot {\left(\frac{N}{\eta }\right)}^{\beta -1}\cdot \exp\left[-{\left(\frac{N}{\eta }\right)}^{\beta }\right];N,\beta ,\eta >0$$
where *β* is the constant scale parameter of the Weibull’s PDF and *η* is its scale parameter, which is dependent on the loading level *S* according to the inverse power-law equation:(4)$$\eta =\eta (S)=10^{a_{0}+a_{1}\cdot \log\left(S\right)};a_{0}>0,a_{1}<0$$

Therefore, the scattered fatigue-life curve that is used to estimate the specimen’s fatigue-life curve and its scatter is modelled with three parameters: *a*_0_, *a*_1_ and *β*. This approach was already successfully applied previously to model the fatigue-life properties of different metallic materials and structures [25,26,27,28,29,30,31,32,33].

However, if this approach is directly applied to materials or products that have a very inhomogeneous and random internal structure or were processed and tested using different procedures the resulting scatter would be enormous and the Weibull’s shape parameter *β* from Equation (3) would be low. One solution to this problem is to model a relationship between the influencing factors (e.g., inhomogeneity types, processing or testing conditions) and the fatigue-life curve model using neural networks. In this way a significant part of the scatter is described by the influencing factors and the fatigue-life curve is determined for a quasi-constant material state or testing conditions. The drawback of this method is that a lot of sample points are needed to obtain a reliable neural network that links the influencing factors to the corresponding fatigue-life curves. In our case, the fatigue-life data for different states of the material and the processing lot of the spruce specimens are rather scarce. For this reason, we decided to augment Equation (4) with additional terms that should reflect the influence of the wood’s density and the processing lot on its fatigue life. This novel model was successfully validated on the spruce specimens that were taken from a serial production of wooden window frames.

The rest of the article is structured as follows. In the second section the static and fatigue experiments are first described, then the experimental data are explained, which is followed by an explanation of the theoretical concepts of the proposed model. In the third section the augmented fatigue-life-curve model is applied to the spruce data. The results are discussed and compared to the static properties of the wood. The article ends with conclusions, acknowledgments and a list of references.

## 2. Materials and Methods

### 2.1. Specimen Preparation

The initial samples were made of Norway spruce (*Picea abies* (L.) *Karst.*) (M SORA d.d., Žiri, Slovenia) and had dimensions of 25 × 25 × 549 mm^3^ (*b* × *h* × *L*) (Figure 1). The samples were produced in two batches (Lot 1 and Lot 2).

In preparing the specimens, special care was taken to ensure that the wood fibres, as well as the growth rings, were aligned as parallel as possible along the entire length of the specimen. The samples were also visually inspected and sorted by their suitable growth-ring alignment and frequency of occurrence in the wood. Prior to testing, the samples were stored for on month under normal conditions at 20 °C and 65% relative humidity. To determine the density *ρ* of each sample, all samples were accurately weighed and measured. The samples to be tested were then selected from our assortment.

### 2.2. Static and Fatigue Experiments

The static three-point bending tests were performed using a Zwick Z100 testing machine (Zwick GmbH ˛Co. KG, Ulm, Germany).

Although the machine can perform many types of tests, we focused on the three-point bending to failure test for our wood specimens. The cylindrical supports were 30 mm in diameter and 350 mm apart, which is 14 times the height of the tested specimens and consistent with ISO 13061-4: 2014 [34], to determine the modulus of elasticity in a static bending test. The load was applied from a cylindrical block of a similar size at the centre of the specimen at a constant rate of 10 mm/min.

The bending tests were performed for two orientations of the wood: the tangential and radial directions.

Two hours after completion of the tests, small pieces (25 × 25 × 20 mm^3^) of the samples were cut to determine the moisture content in all tested samples by drying them in a Kambič SP-210 laboratory dryer (Kambič d.o.o., Semič, Slovenia). The procedure followed the instructions described in the standard ISO 13061-1: 2014 [35]. All measured samples had a moisture content of approximately 11%. The values for the ultimate strength *S*_st_ and the modulus of elasticity *E* were corrected for a 12% moisture content by considering the standards ISO 13061-3: 2014 [36] and ISO 13061-4: 2014 [34]. The results of the static three-point bending tests are shown in Table 1.

Fatigue three-point bending tests were conducted on a self-developed, pneumatic, fatigue test rig (Figure 2). The specimen is supported at the ends by two fixed cylindrical elements with diameters of 30 mm with a separation of 350 mm.

To apply the cyclically pulsating load (dynamic load factor *R* = 0), a pneumatic system, consisting of a compressed-air preparation unit, a control unit, a digital air regulator, two valves and a cylinder, was used (Figure 2). The entire pneumatic system is designed for a maximum loading force of 3500 N at 7 bar. The whole structure of the test rig is described in detail in [37].

The fatigue frequency for the current setup is a 1 s load period followed by a 1 s relaxation period. The fatigue-load data (bending stress amplitude) for each test specimen are given in Table 2. The test was stopped after a pre-set limit for the number of load cycles (2,500,000).

### 2.3. Theoretical Background of the Data Modelling

The basic idea, which was followed when modelling the fatigue-life curve and its scatter for the wood, was to consider the differences between the wood specimens having different density *ρ* values, processing lots and loading directions according to the growth rings. It was observed during the experiments that these three parameters significantly influence the fatigue behaviour of the specimens and can result in different fatigue-life curves. See, for example, Figure 3.

Each of the fatigue-life curves in Figure 3 can be modelled using the conditional Weibull’s PDF *f*(*N*|*S*) from Equation (3) with the constant shape parameter *β* and the load-dependent scale parameter *η*(*S*) following the inverse power-law from Equation (4). To obtain a more universal model of the fatigue-life curves, the functional relationship from Equation (4) is augmented with additional terms to obtain the following multivariate linear-regression model:(5)$$\log\left(\eta \right)=a_{0}+a_{1}\cdot \log\left(S\right)+a_{2}\cdot \rho +a_{31}\cdot D_{1}+a_{32}\cdot D_{2}+\ldots +a_{4}\cdot O;a_{0}>0,a_{1}<0$$

Equation (5) is characterised by a universal intercept coefficient being equal to *a*_0_ and a common slope *a*_1_ for every combination of the processing lot and density *ρ*. The term $a_{2}\cdot \rho$ considers the influence of the wood’s density on the intercept of a particular fatigue-life curve. The wood mass density is an indicator of its dryness. The drier the wood, the lower its mass density should be and the better its mechanical properties should be. Different processing lots are identified through the dummy variables *D*_1_, *D*_2_, etc. Each dummy variable is equal to 1 for the corresponding processing lot. Otherwise, it is equal to 0. Since one processing lot is always used as a reference, the number of dummy variables is equal to the number of processing lots minus one. There are many influencing factors that could be hidden in the different production lots, such as the parameters of the machining process, the origin and history of the logs, etc. In our case, all specimens were machined with the same process parameters, but the other factors were not controlled. This is a common occurrence in practice, as the end user usually does not have complete control over these factors. For this reason, our goal was to assess how different production lots affect the fatigue life. The binary dummy variable *O* represents the orientation of the force during the three-point bending test according to the growth rings. The reference orientation (*O* = 0) is the tangential loading direction. If the loading direction is radial, then *O* is equal to 1. In our case there were two processing lots, so only one dummy variable *D* was used in the model of the Weibull’s scale parameter *η* from Equation (5):(6)$$\eta =\eta \left(S,\rho ,D,O\right)=10^{a_{0}+a_{1}\cdot \log\left(S\right)+a_{2}\cdot \rho +a_{3}\cdot D+a_{4}\cdot O};a_{0}>0,a_{1}<0$$

The dummy variable *D* was equal to 0 for the processing lot 2 and equal to 1 for the processing lot 1. The conditional Weibull’s PDF now has four conditional variables: the loading level *S*, the wood’s density *ρ*, the processing-lot dummy variable *D* and the loading-orientation dummy variable *O*. For this reason it is denoted by *f*(*N*|*S*,*ρ*,*D,O*). Its shape parameter *β* is kept constant. This means that the fatigue-curve model for the spruce specimens is determined using six parameters *a*_0_, *a*_1_, *a*_2_, *a*_3_, *a*_4_ and *β*, which need to be determined from the experimental dataset. The sample set from the fatigue experiments is made up of *n* data points **S** = {(*S_i_*,*N_i_*,*δ_i_*,*ρ**_i_*,*D_i_*,*O_i_*); *i* = 1,…,*n*} with the parameter *δ*_i_ being an indicator of the fatigue failure as follows:(7)$$\delta_{i}=\{\begin{array}{c} 1;\mathrm{fatigue failure}\\ 0;\mathrm{run}-\mathrm{out exp}. \end{array}$$

To properly estimate the six parameters *a*_0_, *a*_1_, *a*_2_, *a*_3_, *a*_4_ and *β* by considering the information that is included in the censored data (i.e., the run-out experiments) the maximum-log-likelihood cost function from Pascual and Meeker is applied [23]:(8)$$L\left(S|\theta \right)=\overset{n}{\underset{i=1}{\sum }}\left\{\delta_{i}\cdot \ln\left[f\left(N_{i}|S_{i},\rho_{i},D_{i},O_{i},\theta \right)\right]+\left(1-\delta_{i}\right)\cdot \ln\left[1-F\left(N_{i}|S_{i},\rho_{i},D_{i},O_{i},\theta \right)\right]\right\}$$
(9)$$\theta =\left(a_{0},a_{1},a_{2},a_{3},a_{4},\beta \right)$$

*F*(*N*|*S*,*ρ*,*D,O*) is the cumulative probability function that corresponds to the Weibull’s PDF *f*(*N*|*S*,*ρ*,*D,O*) with the scale parameter *η* defined in Equation (6):(10)$$F\left(N|S,\rho ,D,O\right)=1-\exp\left[-{\left(\frac{N}{\eta (S,\rho ,D,O)}\right)}^{\beta }\right]$$

Due to the complex relationship between the cost-function value, the six model parameters and the dataset **S**, the maximising of the cost function *L* from Equation (8) was carried out numerically. In our case, a real-valued genetic algorithm was applied for this purpose, which combined the classic single-point crossover and linear crossover of chromosomes during mating as well as a mutation-with-momentum method for mutating the chromosomes. The details of this optimisation algorithm are given in Klemenc and Fajdiga [17] and will not be repeated here.

## 3. Results and Discussion

### 3.1. Static Experiments

The statistical analyses in the continuation were based on the 28 experimental samples in Table 1. They were all calculated with IBM SPSS statistical software. The Pearson’s correlations between the three relevant scale variables (i.e., the spruce’s density *ρ*, the maximum bending stress *S*_max,st_ and the elastic modulus *E* of the tested specimens) are presented in Table 3 for the static experiments.

All the correlations in Table 3 are two-tailed significant at the 0.01 significance level. This indicates the strong dependence of the maximum bending force and the elastic modulus on the spruce’s density. The influence of the two ordinal variables (i.e., the processing lot and the loading orientation) on the maximum bending strength and the elastic modulus were first checked with a series of two-way ANOVA analyses. For the processing lot, the reference was lot 2 (*D* = 0) and for the loading direction, the reference orientation was the tangential direction (*O* = 0). A summary of the results for the four ANOVA analyses is presented in Table 4.

The homogeneity-of-variances tests for the four analyses in Table 4 were relatively insignificant in all the cases. We can conclude that the processing lot has a relatively insignificant influence on both the maximum bending stress and the elastic modulus. The loading direction, on the other hand, has a more pronounced (yet not very significant) influence on the maximum bending stress. Its influence on the spruce’s elastic modulus is relatively insignificant. Based on the results from Table 3 and Table 4, four linear-regression models (LRMs) for the static material properties were set-up and compared:(11)$${\hat{S}}_{max,st}=a_{0,st}+a_{1,st}\cdot \rho +a_{2,st}\cdot O+a_{3,st}\cdot D$$
(12)$${\hat{S}}_{max,st}=a_{0,st}+a_{1,st}\cdot \rho +a_{2,st}\cdot O$$
(13)$$\hat{E}=a_{0,st}+a_{1,st}\cdot \rho +a_{2,st}\cdot O+a_{3,st}\cdot D$$
(14)$$\hat{E}=a_{0,st}+a_{1,st}\cdot \rho$$

Table 5 has a summary for the four LRMs in Equations (11)–(14). To draw conclusions about the applied LRMs, only the model’s quality and the significance of the individual coefficients are meaningful. If the two LRMs from Equations (11) and (12) are compared, it can be observed, based on the *R* and *R*^2^ parameters, that the two models are equivalent in terms of the quality of predicting the spruce’s maximum static bending stress *S*_st_. However, the adjusted *R*^2^ parameter is better when there are fewer independent variables (i.e., in the LRM from Equation (12)) (the adjusted *R*^2^ parameter is an important estimate of the goodness of fit, since it considers also the number of the regression parameters and a size of the sample set beside the agreement of the predicted and measured data). This means that the independent variable *D* (the processing lot) does not have a significant influence on the maximum static bending stress. This is confirmed by the significance of the *a*_3,st_ coefficient in Table 4, which is very close to a value of 1.0 (totally insignificant influence of the independent variable).

A comparison of the LRMs from Equations (13) and (14) in Table 5 leads to the conclusion that the independent variables *O* (the loading direction) and *D* (the processing lot) do not have a significant influence on the elastic modulus *E* of the spruce. Despite the fact that the *R* and *R^2^* parameters for the LRM from Equation (13) are better when compared to the LRM from Equation (14), the corresponding adjusted *R^2^* parameters are the same. As before, the relatively low influence of the *O* and *D* independent variables for the elastic modulus is reflected in the low significance of the corresponding regression coefficients *a*_2,st_ and *a*_3,st_, which are both larger than 0.3. Moreover, the positive regression coefficient *a*_1,st_ in Equations (13) and (14) implies that the higher mass density leads to a higher elastic modulus, which is consistent with the literature data [38,39].

### 3.2. Fatigue Experiments and S-N Curve Model with Scatter

The fatigue-life analyses were performed for the 13 experimental fatigue-life samples in Table 2. To estimate a statistical significance for the different factors that influence the fatigue durability of the spruce specimens, a linear-regression model for the dependent variable log(*N*) (i.e., a logarithm of the number of load cycles to failure) was set up, which resembles the functional form of Equation (5): (15)$$\log\left(N\right)=a_{0}+a_{1}\cdot \log\left(S\right)+a_{2}\cdot \rho +a_{3}\cdot D+a_{4}\cdot O;a_{0}>0,a_{1}<0$$

The parameters as well as the goodness of fit for the LRM in Equation (15) were estimated with the IBM SPSS software. Table 6 has a summary of this LRM. To build this LRM, only the nine samples corresponding to the fatigue-failure samples from Table 2 were considered.

It can be concluded from the results in Table 6 that the logarithm of the loading level log(*S*), the density *ρ* and the processing lot *D* significantly influence the spruce’s fatigue durability. The influence of the loading direction *O* on the dynamic strength of the spruce is less significant when compared to the static strength of the material (see also Table 5). However, if this parameter is omitted from the LRM in Equation (15), the adjusted *R*^2^ parameter of the LRM would be reduced to the value of 0.652 and the significance of the other four regression coefficients would not improve to any extent. This means that the loading direction parameter *O* should also be considered when setting up the S-N curve with scatter model. In contrast to the static characteristics of the material (Table 5), the processing lot *D* has a significant influence on the fatigue-life data.

The S-N curve with scatter was then modelled with a conditional Weibull’s PDF from Equation (3) with the constant shape parameter *b* and the variable scale parameter *h*, which depends on the four independent variables log(*S*), *ρ*, *D* and *O* according to Equation (6). The parameters of this model were estimated using the procedure from Section 2.3. To obtain the best values possible for the six model parameters *a*_0_, *a*_1_, *a*_2_, *a*_3_, *a*_4_, *β* and all the 13 samples in Table 2, ten repetitions of the genetic algorithm were applied with different randomly chosen initial conditions. In each repetition 200,000 iterations of the genetic algorithm were run to ensure convergence to the optimal solution. Our own developed C++ code was used for this purpose. The set-up of the genetic algorithm and the estimated optimal values for the six parameters *a*_0_, *a*_1_, *a*_2_, *a*_3_, *a*_4_ and *β* are listed in Table 7.

In all the repetitions of the genetic algorithm, the final value of the cost function deviated by less than 2% from the maximum value of the cost function for the best solution. Such a consistency of the optimisation processes means that the best solution is near to the global maximum of the cost function in Equation (8). The average value and the standard deviation for the ten repetitions are also listed in Table 7.

If the values for the parameters *a*_0_, *a*_1_, *a*_2_, *a*_3_ and *a*_4_ in Table 7 are compared to the parameters in Table 6, it can be concluded that the model of the fatigue-life curve with its scatter is consistent with the LRM from Table 6. All the coefficients are of a comparable order of magnitude and have the same signs. However, the four run-out samples from Table 2 caused a lower value of the slope parameter *a*_1_ for the model of the fatigue-life curve with scatter. Consequently, the intercept parameter *a*_0_ and the other parameters that position the individual fatigue-life curves in the log(*S*)-log(*N*) space should be different. That is why the absolute values of the four coefficients *a*_0_, *a*_2_, *a*_3_ and *a*_4_ in Table 7 are consistently larger than in Table 6.

If the data in Table 2 are grouped, it can be concluded that there are two groups of data related to the processing lot *D*, two groups related to the loading orientation *O* and approximately three groups of data related to the wood’s density *ρ* (i.e., *ρ* = 400, 440, 485 kg/m^3^). In the continuation, the fatigue-life curves for the different probabilities of fatigue failures are presented in three figures together with the corresponding experimental data points:Figure 4: Fatigue-life curves with scatter for the processing lot 2 (*D* = 0), densities *ρ* = 440 and 485 kg/m^3^ and the tangential loading direction (*O* = 0):Figure 5: Fatigue-life curves with scatter for the processing lot 1 (*D* = 1), densities *ρ* = 400 and 440 kg/m^3^ and the tangential loading direction (*O* = 0);Figure 6: Fatigue-life curves with scatter for the processing lot 1 (*D* = 1), density *ρ* = 400 kg/m^3^ and the radial loading direction (*O* = 1).

In Figure 4, Figure 5 and Figure 6 the dependencies of the maximum stress in the loading cycle *S*_max_ (i.e., the loading level) vs. the corresponding number of load cycles to failure *N* are presented. If the diagrams in these figures are analysed, it can be concluded that the presented model for the fatigue-life curve and their scatter models the experimental data well. In all the figures the experimentally determined data points lie within the scatter band between 10% and 90% probability of failure. This means the comprehensive fatigue-life curve model can also be applied for the case of the radial loading direction even though it is better fitted to the experimental data for the tangential loading direction, which represent more than 80% of the experimental samples. It can be concluded from Table 6 and Table 7, Figure 5 and Figure 6 that a higher density lowers the fatigue-life strength of the spruce, regardless of the processing lot. Such a result is to be expected according to the literature [9,40], since drier wood with lower mass density has higher specific gravity and better mechanical properties. This means that the main influence of mass density on fatigue life of wood was correctly estimated, despite a relatively small amount of experimental data. It can also be concluded from Table 6 and Figure 5 and Figure 6 that the fatigue life of the wood depends on the production lot. Such a result was also expected because the origin of the logs from which the specimens in the two production batches were made was not known. The influence of the loading direction is not significant, which is in agreement with the data in the literature. At a stress level less than half the ultimate bending strength, the test specimens should withstand more than one million loading cycles, which is in accordance with the data in the literature [11]. 

On the other hand, if the density *ρ*, the processing lot *D* and the loading direction *O* were not considered in the fatigue-life curve model and only the slope and intercept parameters *a*_1_ and *a*_0_ were considered in Equation (6), the scatter of the fatigue life curves would be enormous (see Figure 7). The parameters *a*_0_ = 17.975, *a*_1_ = −6.740 and *β* = 0.554 in this case are wrongly estimated (see a comparison with Table 6 or Table 7), because their estimation is made for the whole sample set, without considering the special features of the individual groups of data. It is obvious from Figure 7 that the slope of the fatigue-life curves is too steep, and the scatter is large due to the relatively small value of the Weibull’s shape parameter *β*. This means that it is important that the spruce’s characteristics, like density, processing lot and orientation of the load, with respect to the growth rings are considered when modelling the wood’s strength.

## 4. Conclusions

From the experimental data and the corresponding statistical analyses, it can be concluded that the density of the spruce improves the static bending strength but reduces the fatigue strength. For the static strength, the processing lot is insignificant, although it strongly influences the fatigue resistance of the spruce. Generally, the radial loading direction is more detrimental to the static and fatigue strength than the tangential loading direction, mainly due to delamination effects.

Based on a statistical analysis of the static and the fatigue-life experiments, an augmented model for the fatigue-life curves and their scatter was set-up and validated for the case of bending specimens made of spruce. In the augmented model, the inverse power-law dependency between the loading level *S* and the number of load cycles to failure *N* is supplemented with the effects of the wood’s density, the processing lot and the orientation of the load. The presented model matches the fatigue-life experimental results. It yields a much better fatigue-life prediction than the usual inverse power-law model, which does not consider the special properties of the material and/or the processing history. The main conclusion of this study is that dried wood with the superior specific gravity should be used for demanding building applications where the wood is subjected to fatigue loading. In addition, engineers should be aware that the mechanical properties of the same species of timber, but from different production lots, can vary significantly.

## Figures and Tables

**Figure 1 materials-15-00536-f001:**
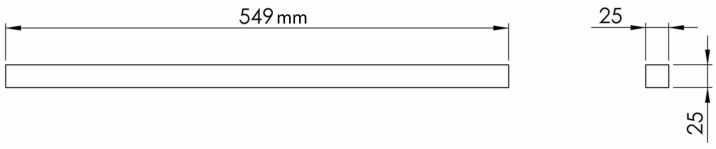
Dimension of a Norway spruce (*Picea abies* (L.) *Karst*.) sample. (unit: mm).

**Figure 2 materials-15-00536-f002:**
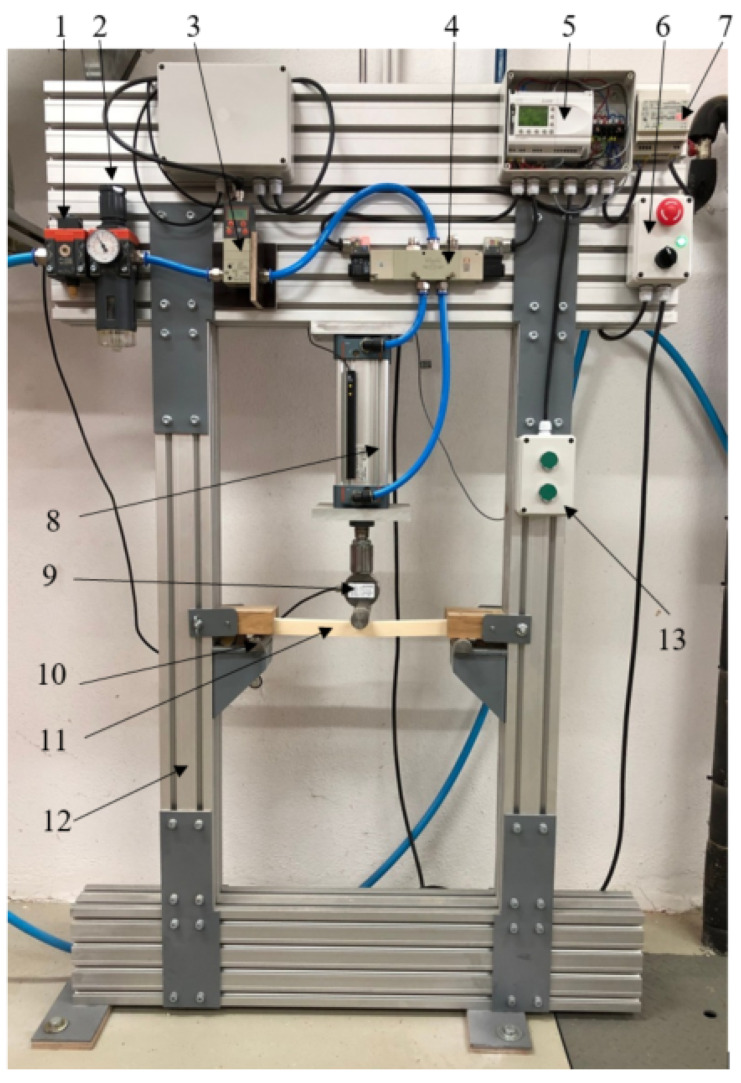
Main fatigue test rig’s components: 1. Cut-off switch, 2. Air preparation unit, 3. Digital air regulator, 4. Two valves, 5. Programmable logic controller, 6. Main power switch (emergency Stop), 7. Power supply 8. Pneumatic cylinder, 9. Load cell, 10. Cylindrical support, 11. Specimen, 12. Aluminium frame, 13. Switches (Start-Stop).

**Figure 3 materials-15-00536-f003:**
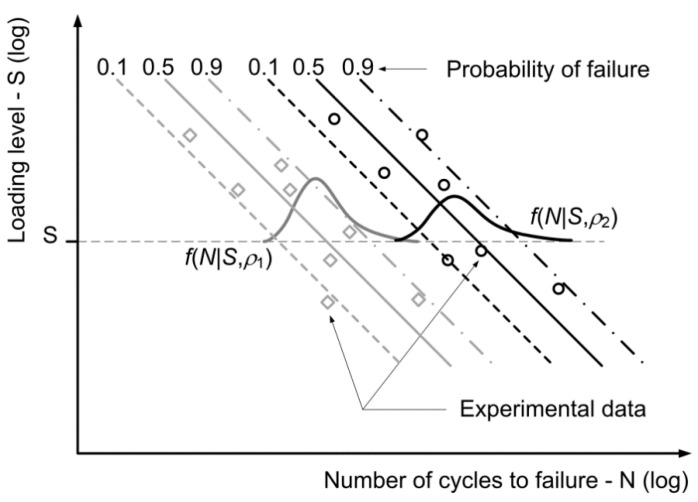
Fatigue-life curves with scatter for different densities of wood.

**Figure 4 materials-15-00536-f004:**
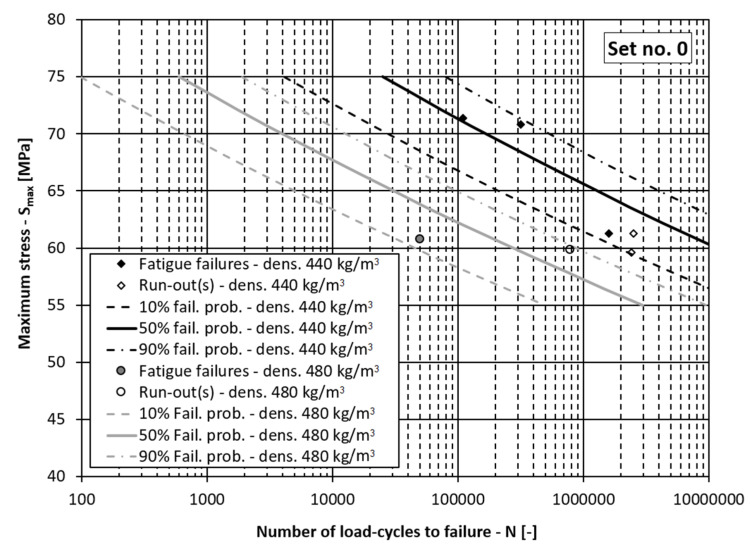
Fatigue-life curves for processing lot 2 and the tangential loading direction. (fail. prob. = failure probability, dens. = wood mass density).

**Figure 5 materials-15-00536-f005:**
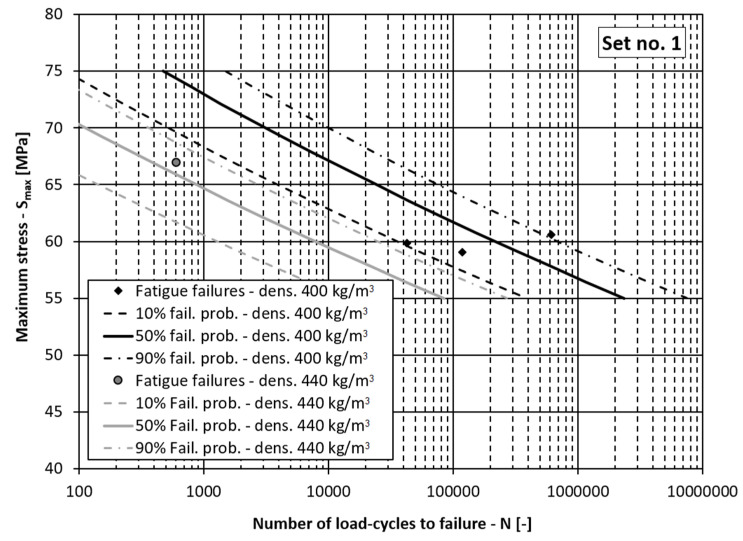
Fatigue-life curves for processing lot 1 and the tangential loading direction. (fail. prob. = failure probability, dens. = wood mass density).

**Figure 6 materials-15-00536-f006:**
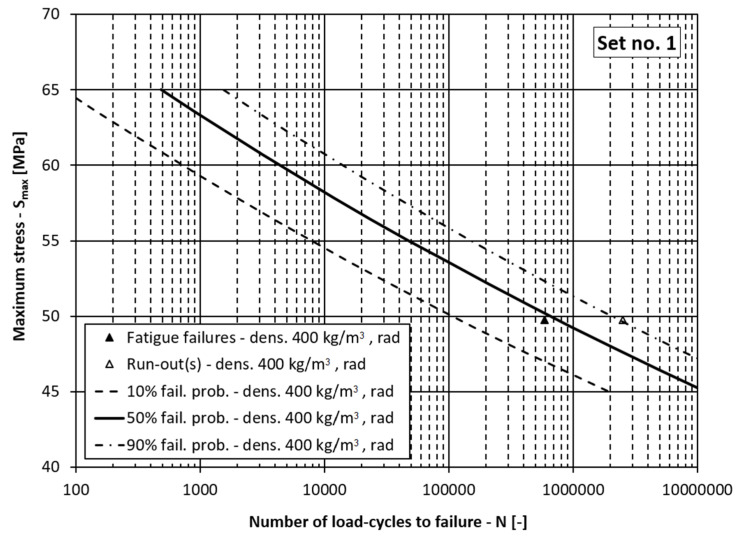
Fatigue-life curves for processing lot 1 and the radial loading direction. (fail. prob. = failure probability, dens. = wood mass density).

**Figure 7 materials-15-00536-f007:**
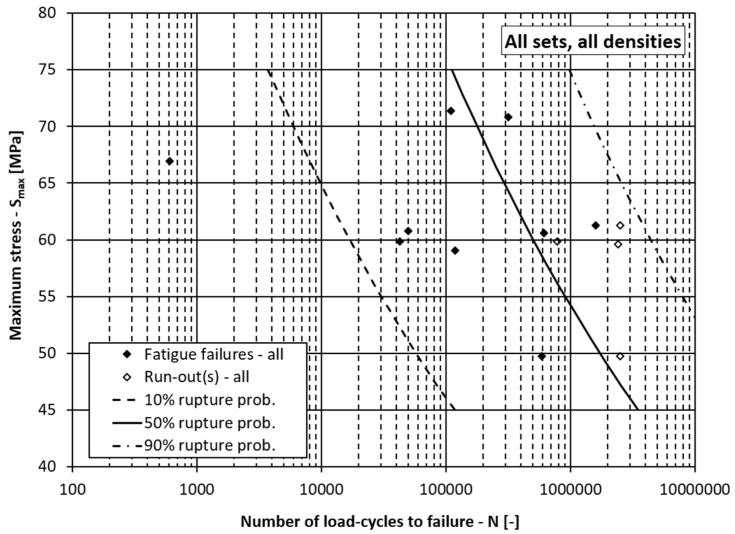
Fatigue-life curves for the summarised data.

**Table 1 materials-15-00536-t001:** Specimen data with the results of the static three-point bending experiments.

Sample Number	Loading Direction	Processing Lot	Density *ρ* [kg/m^3^]	Maximum Force *F* [N]	Maximum Bend. Stress *S*_st_ [MPa]	Elastic Modulus *E* [MPa]
S2	Tangential	Lot 1	540	3372.1	110.2	14,113.4
S22	Tangential	Lot 1	540	3913.0	128.6	14,583.5
S21	Tangential	Lot 1	536	3265.8	106.4	12,743.5
S4	Tangential	Lot 1	509	3450.4	113.4	13,583.2
S14	Tangential	Lot 1	505	2964.5	97.4	12,381.8
S32	Tangential	Lot 1	500	2900.1	94.4	11,899.2
S11	Radial	Lot 1	498	2545.5	83.2	11,051.5
S5	Tangential	Lot 1	491	3284.4	105.2	12,879.9
S33	Radial	Lot 1	489	2865.8	92.9	12,113.3
S3	Tangential	Lot 1	488	3133.2	100.8	12,596.6
S34	Radial	Lot 1	485	2952.2	95.3	12,981.6
S19	Tangential	Lot 1	470	3036.2	99.4	11,412.1
S20	Tangential	Lot 1	467	2997.2	98.1	11,180.5
S13	Radial	Lot 1	464	2587.3	82.9	10,267.6
S24	Radial	Lot 1	464	3002.3	97.3	11,860.6
S12	Tangential	Lot 1	460	2977.1	96.5	12,155.0
S17	Radial	Lot 1	454	2547.4	81.9	10,529.6
S25	Tangential	Lot 1	446	2464.9	79.9	9908.6
S15	Tangential	Lot 1	434	2637.8	86.2	10,226.3
S42	Tangential	Lot 1	422	2548.9	82.7	10,028.9
S40	Tangential	Lot 1	419	2536.4	82.3	9925.6
S35	Tangential	Lot 1	356	2061.1	67.8	7445.9
S55	Tangential	Lot 2	538	3259.7	112.4	14,714.4
S54	Tangential	Lot 2	532	3579.9	124.6	15,168.1
S60	Tangential	Lot 2	460	2355.9	80.9	9960.1
S46	Tangential	Lot 2	455	2705.7	92.7	10,329.3
S57	Tangential	Lot 2	432	2505.1	86.3	11,271.4
S56	Tangential	Lot 2	432	2534.8	86.9	10,965.1

**Table 2 materials-15-00536-t002:** Specimen data with the results of the fatigue three-point bending experiments.

Sample Number	Loading Direction	Processing Lot	Density *ρ* [kg/m^3^]	Bend. Stress Amplitude *S* [MPa]	Cycles to Failure *N* [–]	Fatigue Failure
S8	Tangential	Lot 1	433	66.9	605	Yes
S29	Tangential	Lot 1	402	60.6	614,365	Yes
S9	Tangential	Lot 1	398	59.9	42,838	Yes
S59	Tangential	Lot 2	492	60.8	49,716	Yes
S48	Tangential	Lot 2	479	59.8	782,859	No
S47	Tangential	Lot 2	442	59.6	2,420,000	No
S53	Tangential	Lot 2	442	61.3	1,588,740	Yes
S58	Tangential	Lot 2	440	61.3	2,502,096	No
S45	Tangential	Lot 2	436	70.8	318,222	Yes
S50	Tangential	Lot 2	435	71.4	108,990	Yes
S37	Tangential	Lot 1	398	59.0	118,685	Yes
S27	Radial	Lot 1	391	49.8	589,235	Yes
S39	Radial	Lot 1	386	49.8	2,500,000	No

**Table 3 materials-15-00536-t003:** Correlations between the density, the bending strength and the elastic modulus.

	Wood Mass Density	Max. Bending Stress	Elastic Modulus
Wood mass density	1.000	0.843	0.900
Max. bending stress	0.843	1.000	0.938
Elastic modulus	0.900	0.938	1.000

**Table 4 materials-15-00536-t004:** Results of ANOVA analyses for the bending strength and the elastic modulus.

Analysis No.	Independent Variable	Dependent Variable	F-Statistics	Significance
1	Processing lot	Max. bend. stress	0.158	0.695
2	Loading direction	Max. bend. stress	1.564	0.222
3	Processing lot	Elastic modulus	0.283	0.599
4	Loading direction	Elastic modulus	0.157	0.695

**Table 5 materials-15-00536-t005:** Summary of the LRMs for the static properties of the spruce specimens.

Parameter	LRM—Equation (11)	LRM—Equation (12)	LRM—Equation (13)	LRM—Equation (14)
Correlation coefficient *R*	0.879	0.879	0.908	0.900
Coefficient of determination *R*^2^	0.773	0.773	0.825	0.810
Adjusted *R*^2^	0.745	0.755	0.803	0.803
*a* _0,st_	−32.474	−32.659	−5218.424	−5537.917
Significance of *a*_0,st_	0.043	0.035	0.005	0.002
*a* _1,st_	273.338	273.355	36413.254	36379.343
Significance of *a*_1,st_	0.000	0.000	0.000	0.000
*a* _2,st_	−8.400	−8.467	−290.833	Not applicable
Significance of *a*_2,st_	0.021	0.014	0.446	Not applicable
*a* _3,st_	−0.244	Not applicable	−347.789	Not applicable
Significance of *a*_3,st_	0.943	Not applicable	0.363	Not applicable

**Table 6 materials-15-00536-t006:** Summary of the LRM for the fatigue life of the spruce specimens.

Parameter	Goodness of Fit and LRM Parameters	Significance of Regression Coefficients
Corr. coeff. *R*	0.920	Not applicable
Coeff. of det. *R*^2^	0.847	Not applicable
Adjusted *R*^2^	0.694	Not applicable
*a* _0_	57.640	0.017
*a* _1_	−20.103	0.049
*a* _2_	−34.831	0.027
*a* _3_	−2.902	0.012
*a* _4_	−1.234	0.263

**Table 7 materials-15-00536-t007:** Summary of the LRM for the fatigue life of the spruce specimens.

Parameter for GA Settings		Value	
Population size		20	
Probability of cross-over		0.6	
Fraction of linear cross-over		0.5	
Probability of mutation		0.05	
Moment weight		1.0	
Moment threshold		1.0	
Estimated parameters	Best solution	Average value	Std. deviation
Final value of the cost function	−121.147	−121.917	0.566
*a* _0_	71.879	65.404	12.910
*a* _1_	−27.441	−25.084	5.402
*a* _2_	−36.071	−31.173	8.692
*a* _3_	−3.169	−2.867	0.518
*a* _4_	−1.693	−1.420	0.632
*β*	1.043	0.988	0.048

## Data Availability

The data presented in this study are available on request from the corresponding author.

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
