# Peer review of "Statistical Modelling of the Fatigue Bending Strength of Norway Spruce Wood"

_materials, 2022, doi:10.3390/ma15020536_

Round 1

Reviewer 1 Report

The objective of this manuscript was to develop a novel approach for modelling the fatigue-life properties of wood based on a statistical analysis of the static and the fatigue-life experiments.

The title is appropriate. The novelty of this paper does not rely on the use of linear-regression models (LRMs) to bending stress and modulus of elasticity considering as independent variables the Processing lot, Loading direction and density, but mostly to fatigue data. The behaviour in fatigue of spruce wood has not been much addressed in the literature.  

The state-of-the-art section gives an overview of the works published in the literature about the statistics methodology applied for data analysis of this kind of fatigue data.

In materials and methods, the methods for static bending and fatigue three-point bending tests were well described, as well as the equations of Weibull’s probability density function.

The conclusions are well supported by the results.

Some minor revisions are also indicated in the manuscript. So, I think that the paper should be accepted with minor revisions.

Reviewer 2 Report

In this paper a novel approach to modelling the fatigue-life properties of Norwegian spruce wood (Picea abies) was presented. The topic is in current interests for the readers. Authors have to add some changes and additions in the text.

Introduction

  • The state of the art has been correctly presented. Authors have to explain why they cited their own 14 papers (in total 39) - that means more than 30%. Is the topic popular only in Slovenia?

Materials and Methods

  • There is a lack of the properties of the Norwegian spruce wood and preparation of samples used for the investigations. Such information is necessary from the "wood science" point of wood and useful for deeper analysis.
  • How long samples were stored in the laboratory?
  • In the Tables 1-2 the density in another unit (kg/m3) should be given.
  • Authors wrote "... Two hours after the tests were completed, small pieces of samples were cut to determine the content of moisture ..." Authors should add information about amount and dimensions of samples to the moisture content determination.

Results and Discussion

  • Authors should consider the use of the density values in the kg/m3. and describing relationships in figures with formulas.

Conclusions

  • Authors wrote "... In addition, engineers should be aware that the mechanical properties of the same species of timber, but from different production lots, can vary significantly ..." Taking this into account, more information on the properties of wood samples used in the experiments should be provided in the Materials and Methods section.

References

  • Authors cited 39 papers and standards. It can be found more than 1/3 papers of the one co-Author Mr. Gorazd Fajdiga. Authors should explain it and made changes in the text (e.g. delete some papers) .

I recommend the paper for the publishing after minor changes and additions.

Reviewer 3 Report

Dear Authors,

It is an interesting manuscript expanding the knowledge of the mechanical properties of Norway spruce wood as an important structural biomaterial. In my review, I will focus on a few important elements that need to be improved or completed. I present the suggestions for additions and corrections in a synthetic way.  

Title – lines: 1- 2
The title should be corrected. My proposition:

Statistical modeling of the fatigue bending strength of Norway spruce wood

Durability is a more appropriate word for studying the behavior of wood under the influence of biotic factors. The title should clearly specify the species of the tested wood.

Introduction – lines: 28 – 134
In the chapter "Introduction" as many as 34 references were cited, of which 50% (17 articles) are self-citations. These are not the correct proportions. This impression is deepened by the analysis of the content of self-cited works. Some of them are thematically little related to the leading theme of the reviewed manuscript.

I propose to do a new literature search and cite more works by other authors on the fatigue strength of wood and reduce the number of self-citations.

Materials and Methods
Specimen preparation  – line 137

The correct name of the tested wood should be given: Norway spruce (Picea abies (L.) Karst.).
The standard Latin name of Norway spruce wood is Picea abies (L.) Karst. according to EN 13556:2003 Round and sawn timber – Nomenclature of timbers used in Europe.

Figure 1 – between 139 and 140
I suggest removing both photos: Figure 1a and 1b and in this place inserting a technical drawing with a dimensioned sample (in accordance with the technical drawing rules).

In addition, Figure 1b is a plagiarism drawing used in the article: Experimental and numerical determination of the mechanical properties of spruce wood – doi:10.3390/f10121140

The same problem concerns Figure 2 and text in lines 165-171 (they are identical to the article: Experimental and numerical determination of the mechanical properties of spruce wood – doi:10.3390/f10121140).

Figure 2 should be removed. Figure 2 is not needed at all, because all relevant information is presented and selected in Figure 3.

Table 1  - between 180 and 181 line
Data from Table 1 repeat with the results presented in the article: (Experimental and numerical determination of the mechanical properties of spruce wood – doi:10.3390/f10121140). I suggest removing table 1 and in this place to give a link to the results presented in the earlier article - simply cite this article.

Table 2 –between 192 and 193 line
I suggest changing wood density units from ton/m3 per kg/m3.

Results and Discussion
The analysis and discussion of results are substantively correct in my opinion.
However, the obtained research results should be more broadly related to other articles in this field.

In addition, the text edits should be improved: lines 282-285, 294-295.

Yours sincerely
Reviewer

Reviewer 4 Report

The authors should discuss why they are interested in investigating fatigue in wood structures.

It does not seem that wood structures fail due to fatigue loadings but most of their reliability is due to environmental actions such as harsh environment which reduce stiffness and strength but failure does not occur due to fatigue.

Results agree well with the expectations and parameters provided give help to designers in order to employ such fatigue life curves for structural design.

Introduction could be improved by including recent studies about wood structures and wood technologies. Some suggestions are provided:

-- https://doi.org/10.1016/j.conbuildmat.2021.124457

-- https://doi.org/10.1016/j.compstruct.2021.114319

-- https://doi.org/10.1016/j.compstruct.2020.112736

Round 2

Reviewer 3 Report

Most of the comments from my review on 12/19/2021 have been applied to the revised version of the manuscript. There are still a few minor editing imperfections to be corrected. I present them in a synthetic way.

Lines – 486 – 491
The most important is to improve the numbering of the cited literature. The numbers 31, 32, 34, and 35 have been omitted from the numbering list.  As a result, the numbers of works 33 and works from 36 to 43 will change. The numbering of references in the text should also be corrected: lines: 110, 154, 162, 165, 180, 297, 377, and 386.

Line 352
It is: =.440 and it should be: = 440

Line 425
It is: (WOOLF-OP20.03520.  and it should be: (WOOLF-OP20.03520).  

Line 450
I propose to complete the bibliographic data:
United States Department of Agriculture Forest Service. Forest Product Laboratory.
